# Open-Cell Spray Polyurethane Foams Based on Biopolyols from Fruit Seed Oils

**DOI:** 10.3390/polym16081145

**Published:** 2024-04-19

**Authors:** Maria Kurańska, Elżbieta Malewska, Hubert Ożóg, Julia Sędzimir, Aleksandra Put, Natalia Kowalik, Sławomir Michałowski, Marcin Zemła, Michał Kucała, Aleksander Prociak

**Affiliations:** Department of Chemistry and Technology of Polymers, Cracow University of Technology, Warszawska 24, 31-155 Cracow, Poland; elzbieta.malewska@pk.edu.pl (E.M.); hubert.ozog@student.pk.edu.pl (H.O.); julia.sedzimir@student.pk.edu.pl (J.S.); aleksandra.put@student.pk.edu.pl (A.P.); natalia.kowalik@student.pk.edu.pl (N.K.); slawomir.michalowski@pk.edu.pl (S.M.); marcin.zemla@doktorant.pk.edu.pl (M.Z.); michal.kucala@doktorant.pk.edu.pl (M.K.); aleksander.prociak@pk.edu.pl (A.P.)

**Keywords:** spray biofoam, open-cell foam, biopolyol, thermal insulation, fruit seed oil

## Abstract

Natural oils from watermelon, cherry, black currant, grape and pomegranate fruit seeds were applied in the synthesis of biopolyols using the transesterification reaction. In this manuscript, the preparation possibility of open-cell foams from a polyurethane system in which petrochemical polyol was fully replaced with biopolyols is analyzed. Firstly, polyurethane foam systems were developed on a laboratory scale, and they were next tested under industrial conditions. It was shown that the foaming method has a significant impact on the foaming process and the cell structure of obtained foams as well as their thermal insulation properties. Based on the conducted research, it was found that the method of processing the polyurethane system has a significant impact on the properties of open-cell spray foams. Foams produced under industrial conditions have a much higher cell density, which has a positive effect on their selected physical–mechanical properties compared to foams produced on a laboratory scale. The open-cell biofoams obtained using a high-pressure machine had apparent densities 12–17 kg/m^3^, thermal conductivity coefficients 35–37 mW/m·K, closed-cell contents < 10% and were dimensionally stable at low and high temperatures.

## 1. Introduction

The new challenges related to reducing greenhouse gas emissions in connection with the European Green Deal motivate scientists to look for new ecological solutions in the field of polymer materials [1,2,3,4]. Thermal insulation materials play an important role in reducing energy losses due to their excellent thermal insulation properties. In addition, thermal insulation polyurethane foams can be partially obtained from renewable raw materials.

The energy used for heating or cooling in the construction sector, according to literature data, is 40%. For this reason, there is a growing interest in energy-efficient buildings [5]. Closed- and open-cell polyurethane (PUR) foams can be used as efficient thermal insulation materials. Closed-cell PUR foams are applied for the hydro and thermal insulation of the roofs, walls and foundations of buildings. The foams of this type have apparent densities of 30–60 kg/m^3^, closed-cell contents higher than 90%, thermal conductivity coefficients up to 28 mW/m·K and water absorption lower than 2%. In the case of open-cell PUR foams, the apparent density is significantly lower at 8–15 kg/m^3^, with closed-cell contents lower than 20%, which influences thermal conductivity (35–42 mW/m·K), which is higher in comparison to closed-cell foams [6].

Currently, on the industrial scale, the raw materials for synthesis of PUR foams are obtained from petrochemical sources, which contribute to increased greenhouse emissions. The possibilities of replacing petrochemical polyols with biopolyols from renewable raw materials mainly concern PUR foams with a closed-cell structure. The most frequently used renewable or waste raw materials for the production of polyols are vegetable oils [7,8], lignin [9,10,11] and polymer waste [12,13]. In order to synthesize biopolyols from renewable sources, chemical modification or biological conversion is necessary depending on the type of the starting material chosen and the expected properties of the final products. The methods of vegetable oil modification are almost always based on the carbon–carbon double bonds or ester groups of triglycerides, involving hydroformylation, transesterification and ozonolysis followed by hydrogenation and epoxidation with epoxy ring opening [14].

Paciorek-Sadowska et al. obtained a new biopolyol based on Oenothera seed oil in a two-step method of epoxidation of double bonds and opening epoxy rings with 2,2′-mercaptodiethanol. This biopolyol was used as a raw material for the synthesis of closed-cell polyurethane-polyisocyanurate foams [15,16]. A future of using edible oils as a major alternative to petrochemical polyol may pose a critical issue of competition with food supplies. It is also possible to use non-edible vegetable oils such as Karanja oil [13], tung oil [17,18], tall oil [19,20,21,22,23] and cardanol [24,25]. Luo et al. reported the feasibility of using crude glycerol in biopolyol synthesis and the production of PUR foams with apparent density of 43 kg/m^3^. Kosmela et al. applied biomass liquefaction to produce biopolyols suitable for the manufacturing of rigid polyurethane foams with closed-cell structure and apparent density ca. 40 kg/m^3^ [26]. Zhang et al. modified PUR foams with four kinds of agricultural waste (oilseed rape straw, rice straw, wheat straw and corn stover) liquefied polyols [27].

To the best of our knowledge, there are few studies that have been reported to use modified vegetable oils to substitute petrochemical polyols in the formulation of open-cell PUR foams. The literature is particularly limited when it comes to the preparation of open-cell foams with low apparent densities (<20 kg/m^3^). In our previous work, we analyzed the influence of the biopolyols from oils such as cooking oil, hemp oil and radish oil on the properties of open-cell PUR foams [28,29]. The properties of the obtained foams were comparable to commercial materials [6]. In order to search for new sources of raw materials for the production of polyols, oils from the waste seeds of various fruits were analyzed in terms of their suitability in the synthesis of biopolyols.

## 2. Materials and Methods

### 2.1. Materials

Blackcurrant (BR), cherry (CH), grape (GR), pomegranate (PG) and watermelon (WM) seed oils were purchased from OlVita, Poland. In the transesterification reaction, triethanolamine (TEA) (Avantor Performance Materials Poland, Gliwice, Poland) as a transesterification agent and anhydrous zinc acetate (Chempur, Piekary Śląskie, Poland) as a catalyst were used. In order to obtain open-cell spray PUR foams, various raw materials were used. The catalyst (POLYCAT^®^142—1,1,3,3-Tetramethylguanidine, non-emissive, highly efficient reactive amine promoting the blowing reaction with a quick initiation) and surfactants (TEGOSTAB^®^ B 8870—polyether-polysiloxane copolymer, a strong stabilizer used in high-water formulations; TEGOSTAB^®^ B 8523—polyether-polydimethylsiloxan copolymer, a strong cell opener; and ORTEGOL^®^ 500—a strong silicone-free cell opener) were provided by Evonik Industries AG (Essen, Germany). Flame retardant tris(2-chloroisopropyl)phosphate (TCPP) with a viscosity of 67 mPa·s at 25 °C was supplied by Purinova (Bydgoszcz, Poland). The isocyanate pMDI Purocyn B with an isocyanate group content of 31% was supplied by Purinova (Bydgoszcz, Poland). Distilled water played the role of chemical blowing agent.

### 2.2. Synthesis of Biopolyols

Biopolyols were synthesized using the transesterification method for fruit seed oils with triethanolamine (TEA). The reaction was carried out in a 6 dm^3^ reactor equipped with a mechanical stirrer, at atmospheric pressure under reflux conditions, and in an inert gas (nitrogen) atmosphere. The molar ratio of the transesterification agent—TEA to used oils was 3:1 and 0.45 wt.% related to the oil mass. The reaction was conducted at 175 °C for 2 h. The obtained biopolyols were characterized by the following hydroxyl numbers and viscosity: the biopolyol from blackcurrant seed oil (BR)—380 mgKOH/g and 180 mPa·s; the biopolyol from cherry seed oil (CH)—370 mgKOH/g and 210 mPa·s; the biopolyol from grape seed oil (GR)—380 mgKOH/g and 150 mPa·s; the biopolyol from pomegranate seed oil (PG)—360 g/mol and 530 mPa·s; the biopolyol from watermelon seed oil—360 mgKOH/g and 190 mPa·s.

### 2.3. Preparation of Open-Cell PUR Foams on a Laboratory and an Industrial Scale

Open-cell PUR foams were prepared using a one-step method with a two-component system. Component A consisted of a biopolyol, catalyst, surfactant, flame retardant and water, while component B was isocyanate. Component A was mixed for 30 s. The temperature of components A and B was 30 °C in order to simulate standard PUR foam production conditions using a high-pressure spray machine. After reaching the desired temperature, component B was rapidly poured into component A, and the mixture was stirred for approximately 4 s before being poured into an open vertical mold which provided free growth of the foam (Figure 1). Each PUR system was colored a specific color to easily identify which system was being sprayed under industrial conditions. The volume ratio of Component A to Component B was 1:1.

The open-cell PUR foams developed on a laboratory scale were obtained on an industrial scale using a high-pressure spraying machine, a Reactor E-20 made by Graco Inc. (Minneapolis, MN, USA) equipped with a Graco Fusion^®^ AP spray gun. The temperature of the components at the mixing head was 30 °C. The spraying pressure was 100 bar. The ambient and sprayed surface temperature was 10 °C. Spraying was carried out on horizontally placed cardboard (Figure 2).

Samples were cut out 24 h after obtaining the foams. The names of the samples come from the type of material (PUR), the type of biopolyol (BR, CH, GR, PG, WM) and the synthesis method (lab—laboratory scale, or ind—industrial scale). The formulations of the PUR systems are shown in Table 1.

### 2.4. Methods of Biopolyol Characterization

Testing the water content using Karl Fischer’s method was performed according to the PN-81/C-04959 standard [30] with the use of a TitroLine KF device manufactured by SCHOTT Instruments GmbH (Mainz, Germany). The determination of the hydroxyl value of the polyols was performed following the non-pyridine method, which is based on the acetylation reaction of OH groups of a polyol with pyromellitic dianhydride in an acetone medium using 1-methylimidazole as a catalyst. The excess anhydride decomposes with water and the resulting acid is titrated with an NaOH solution in the presence of an indicator. The viscosity was determined with a HAAKE MARS III rotary rheometer from Thermo Scientific (Waltham, MA, USA). During the measurements, a plate-to-plate system at 100 rpm was used. This study was conducted at 25 °C.

### 2.5. Methods of PUR Foam Characterization

The apparent densities of the foams were determined on the basis of the measurements of the samples’ masses and volumes according to the ISO 845 standard [31]. The closed-cell content was studied according to the ISO 4590 standard [32]. Foam morphology was analyzed using optical microscope images. The cell morphology was analyzed using an optical microscope (PZO Warszawa, Warszawa, Praga). All samples for structure tests were taken from the center of the foam, the so-called core. The images were taken in cross-sections perpendicular to the direction of foam growth. Cell density was calculated using Equation (1):(1)$$N={\left(\frac{n}{A}\right)}^{\frac{3}{2}}$$
where *N* is the cell density expressed as the number of cells per cm^3^, and *n* is the number of cells per area *A* (in cm^2^).

The heat conduction coefficient (λ) of the PUR foams was determined using a Laser-Comp FOX 200 apparatus from TA Instruments (New Castle, DE, USA). The temperature difference between the hot and cold plate was 20 °C. The compressive strength test was carried out in accordance with the ISO 844 standard [33] using a Zwick Z005 TH testing machine (Zwick GmbH & Co., Ulm, Germany). The compressive strength was determined parallel to the direction of foam growth at 10% deformation. The dimensional stability (DS) was tested according to the ISO 2796-1986 standard [34]. The dimensional stability was calculated using Equation (2).
(2)$$DS_{l}=\left(\frac{l_{f}-l_{i}}{l_{i}}\right)\times 100,\;DS_{w}=\left(\frac{w_{f}-w_{i}}{w_{i}}\right)\times 100,DS_{t}=\left(\frac{t-t_{i}}{t_{i}}\right)\times 100$$
where *DS* is the dimensional stability, *l_i_* is the initial length, *l_f_* is the length after thermal treatment, *w_i_* is the initial width, *w_f_* is the width after thermal treatment, *t_i_* is the initial thickness, and *t_f_* is the thickness after thermal treatment.

The limiting oxygen index (LOI) test was performed on the basis of the ISO 4589-2 standard [35].

## 3. Results and Discussion

The development of PUR systems processed by spraying requires a different approach compared to typical PUR foams. Typically, the isocyanate index of foams described in the literature is approximately 100–110. In the case of spray foams, this index is much lower ca. 50 due to the need to ensure a 1:1 volumetric dosage during spraying. In order to analyze the influence of the processing method of PUR systems on foaming process and foam properties, foams were made using a mechanical stirrer in the lab and an industrial high-pressure spray device. At the first stage, the impact of the type of processing of the PUR systems on the apparent density of the obtained foams was assessed, in which 100% of the petrochemical polyol was replaced with the biopolyols from blackcurrant, cherry, grape, pomegranate and watermelon seed oils. The influence of the biopolyol type and of the PUR system processing method on the apparent density of the obtained foams is shown in Figure 3.

In the case of open-cell PUR foams obtained on a laboratory scale, there was no significant effect of the type of biopolyol on the apparent density of the foams. All materials had a similar apparent density ca. 10.5 kg/m^3^. The PUR_PG_lab foam had a slightly higher apparent density of approximately 12 kg/m^3^. This effect may be related to the higher viscosity of the biopolyol PG, which may limit expansion when obtaining foams in laboratory conditions. The opposite effect was observed in the case of material production under industrial conditions using a high-pressure device. In this case, the biofoams had more diverse apparent densities. Among the materials produced under industrial conditions, the foams had apparent densities in the range from 11.8 to 17.0 kg/m^3^. The foam obtained using the biopolyol from watermelon seed oil unexpectedly was characterized by the highest apparent density.

Polaczek and Kurańska obtained biofoams based on biopolyols from vegetable oils: hemp seed oil, oilseed radish oil, rapeseed oil and used rapeseed cooking oil. The foams prepared on a laboratory scale were characterized by apparent densities ranging from 11.2 to 12.1 kg/m^3^. In their work, the isocyanate index was [29]. In another work on the synthesis of biofoams using laboratory and industrial methods, it was also found that foams obtained using the industrial method had higher apparent densities. The apparent densities of the foams ranged from approximately 14 kg/m^3^ to over 21 kg/m^3^. All of the open-cell polyurethane foams had higher apparent densities compared to the foams synthesized in the laboratory with the use of a mechanical mixer [6].

The processing method of PUR systems has a significant impact on the cellular structure of obtained porous materials. The cellular structure has a decisive influence on both the thermal insulation and mechanical properties of open-cell PUR foams with low apparent density. Micrographs of the cellular structures of the foams obtained under laboratory and industrial conditions are shown in Figure 4. Figure 5 presents their cell densities as well as the contents of closed cells.

The use of a high-pressure spray device resulted in a significant reduction in cell size, which is shown in Figure 4 and is reflected in the cell density of foams obtained in industrial conditions. A similar effect was obtained by Kurańska et al. when increasing the scale of production of closed-cell polyurethane biofoams. The authors found that the use of a high-pressure device allows for reducing the cell size of the resultant foams compared to the foams produced on a laboratory scale [36]. It was confirmed that the reduction of the cell diameter reduced the thermal conductivity of open-cell foams. According to the principles of heat transport in porous materials, as cell size decreases, free gas convection and heat flow through radiation (it has a small share in the total heat flow) in open-cell PUR foams decrease. The influence of the processing method of biofoams on their thermal conductivity is shown in Figure 6. It was confirmed that the better cell structure of the foams obtained under industrial conditions is reflected in better heat insulating properties.

All of the foams, regardless of the biopolyol type used and the processing method, had an open cell structure because the content of closed cells was below 10% (Figure 5). Foams obtained under laboratory conditions had a thermal conductivity coefficient in the range of 0.038–0.043 W/(m·K), while in the case of foams obtained under industrial conditions, values of this parameter were in the range of 0.035–0.038 W/(m·K). Such a range is slightly more favorable compared to the foams obtained from petrochemical raw materials on an industrial scale. The thermal conductivity of open-cell PUR foams in industry is in the range 0.037–0.039 W/(m·K).

The mechanical properties of the foams were investigated in two directions: parallel and perpendicular to the foam rise direction. This is due to cell anisotropy, which causes different mechanical properties of foams measured in the directions parallel and perpendicular to the growth of the foam. The compressive strength values in the parallel direction are higher compared to the perpendicular direction, which is a consequence of the anisotropic character of the obtained foams. Compressive strength values depending on the type of biopolyol from which the foams were made and the processing methods are shown in Figure 7.

Foams obtained under industrial conditions are characterized by higher compressive strength. This effect is caused by two factors. The first factor is the higher apparent density of foams obtained under industrial conditions despite the use of the same foam recipe. The second factor is the finer cell structure of these foams, which has a beneficial effect on the mechanical properties of foams. As is clearly stated in the literature, the mechanical properties of porous polyurethane are closely correlated to their apparent density due to an increase in the cell wall thickness and/or a decrease in the cell size; according to Euler–Bernoulli beam theory, this gives rise to an increase in the bending moment of the cell wall [37]. And this correlation can be clearly noticed for foams prepared in industrial conditions. Foams with higher apparent density exhibit higher compressive strength.

Commercial foams are characterized by a minimum mechanical strength of approximately 10 kPa and the value is usually given in one direction of the test. Regardless of the production method, all foams are characterized by a strength higher than 10 kPa analyzed in the direction parallel to the direction of growth.

By comparing foam materials made from different biopolyols, it can be concluded that a wide range of vegetable oils can be used to produce polyurethane foams. Natural oils from watermelon, cherry, black currant, grape and pomegranate fruit seeds are a very valuable raw material for obtaining hydroxyl derivatives as starting material ingredients in polyurethane systems. The properties of the foams obtained in industrial tests are comparable and it is not possible to single out the best biopolyol. However, in the case of sprayed polyurethane foams, it is important to synchronize the characteristic foaming times to prevent the polyurethane system from flowing from the wall/ceiling. Systems developed in this research based on different biopolyols had the same catalytic system. Figure 8 shows the differences in the foaming process on the vertical surface.

In the case of the PUR_WM_ind system, the PUR system flows off the wall (marked with a blue frame), which is an unfavorable phenomenon because the foam should immediately expand, as in the case of the PUR_PG_ind system. A similar effect as in the case of PUR_WM_ind was observed for PUR_BC_ind, PUR_CH_ind and PUR_GR_ind systems. This effect can be offset by the appropriate selection of catalysts. The properties of biopolyols, specifically their viscosity, turned out to be important in these studies. The PUR_PG_ind system was obtained from the biopolyol with the highest viscosity (530 mPa·s) and this could have resulted in a faster expansion process on the vertical wall.

Despite the low values of the apparent density and compressive strength of the foams, the results obtained for the dimensional stability at elevated (70 °C) and low (−25 °C) temperatures were satisfactory (Table 2).

The changes in the dimensions of the samples after conditioning in different temperatures for 24 h were lower than 1% regardless of the measurement direction. In the case of the open-cell foams, the difference between the internal cell gas pressure within the foam and the external atmospheric pressure was zero. Thanks to this, the dimensional stability is maintained and open-cell foams can have a very low apparent density without shrinkage problems. Such a property makes them suitable for application as thermal insulation materials.

Measuring the oxygen index (LOI) is one of many basic methods for assessing the flammability of polymeric materials. The oxygen index of the PUR biofoams obtained under laboratory and industrial conditions is presented in Table 2. The LOI values above 21% indicate that the PUR foam will not support the combustion process in air. All the foams obtained under industrial conditions, regardless of the type of biopolyol used for their synthesis, were characterized by an oxygen index above 21%. In the case of the foams obtained under laboratory conditions, the LOI values were slightly lower, which may be related to the lower apparent density of the materials. The biofoams developed as part of this research contain flame retardants in their structure. The higher the apparent density, the more solid polyurethane per volume unit, which results in lower flammability.

## 4. Conclusions

New biopolyols based on black currant, cherry, grape, pomegranate and watermelon fruit seeds were obtained in transesterification reaction with triethanolamine. These biopolyols were used as the raw material for the synthesis of open-cell polyurethane foams replacing 100% petrochemical polyol. The results confirm that a biopolyol based on used cooking oil can be successfully applied in the preparation of open-cell polyurethane foams. The foams developed on a laboratory scale were then obtained under industrial conditions. It was found that the conditions in which the materials were produced have a significant impact on the properties of the obtained foams. The biofoams obtained on a laboratory scale were characterized by apparent densities in the range of 10–12 kg/m^3^, while the foams obtained in industrial conditions had apparent densities in the range of 12–17 kg/m^3^. The effect of the foaming conditions was particularly visible in the case of the cellular structure of the foams. Foaming polyurethane systems under laboratory conditions resulted in obtaining foams with much larger cell sizes compared to the foams obtained under industrial conditions. This had a direct impact on the values of the heat conduction coefficient. The thermal conductivity coefficient for the foams obtained on a laboratory scale was in the range of 38–43 mW/m·K, while for the open-cell polyurethane foams obtained in industrial conditions it was 36–38 mW/m·K and comparable to that found for commercial foams. Based on the relationship between the thermal conductivity and the cell size of the open-cell biofoam samples, smaller cell size improved the thermal insulation property of the materials. The results confirm that it is possible to obtain open-cell polyurethane foams from fruit seed oils with favorable functional properties comparable to those of typical commercial foams.

## Figures and Tables

**Figure 1 polymers-16-01145-f001:**
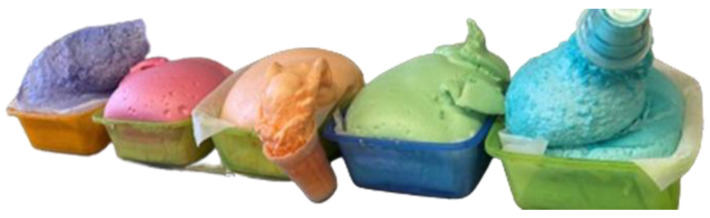
Open-cell PUR foams prepared in laboratory (lab).

**Figure 2 polymers-16-01145-f002:**
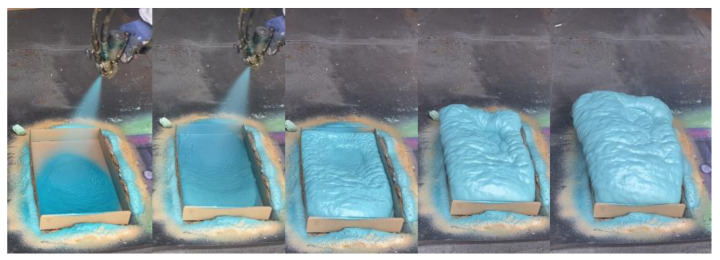
Open-cell PUR foams prepared under industrial conditions (ind).

**Figure 3 polymers-16-01145-f003:**
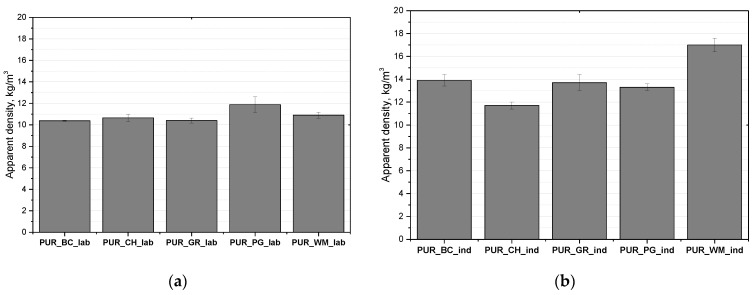
Apparent densities of open-cell PUR biofoams obtained under laboratory (**a**) and industrial (**b**) conditions.

**Figure 4 polymers-16-01145-f004:**
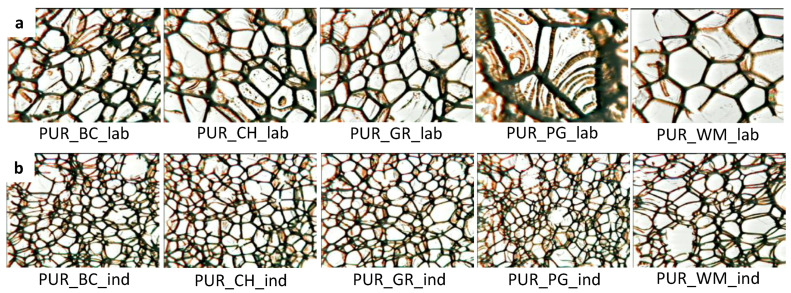
Cellular structures of open-cell PUR biofoams obtained under laboratory (**a**) and industrial (**b**) conditions.

**Figure 5 polymers-16-01145-f005:**
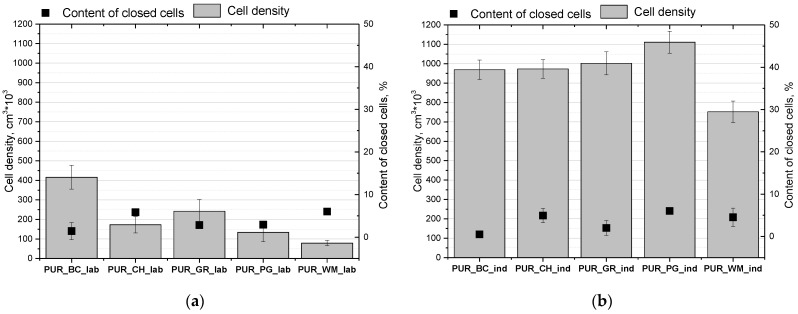
Cell densities and the contents of closed cells of open-cell PUR biofoams obtained under laboratory (**a**) and industrial (**b**) conditions.

**Figure 6 polymers-16-01145-f006:**
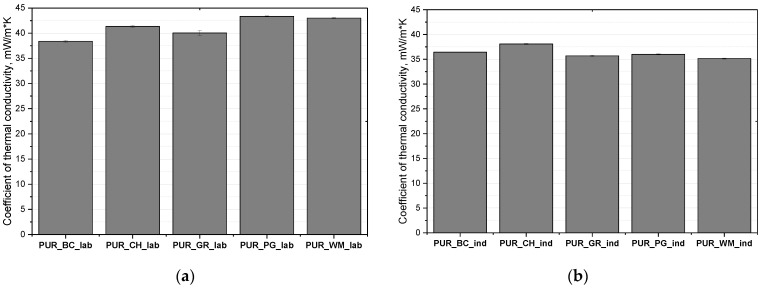
Thermal conductivity coefficients of open-cell PUR biofoams obtained under laboratory (**a**) and industrial (**b**) conditions.

**Figure 7 polymers-16-01145-f007:**
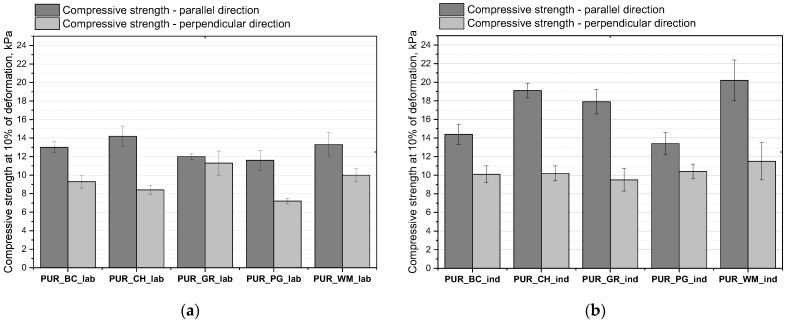
Compressive strength of PUR biofoams obtained under laboratory (**a**) and industrial (**b**) conditions.

**Figure 8 polymers-16-01145-f008:**
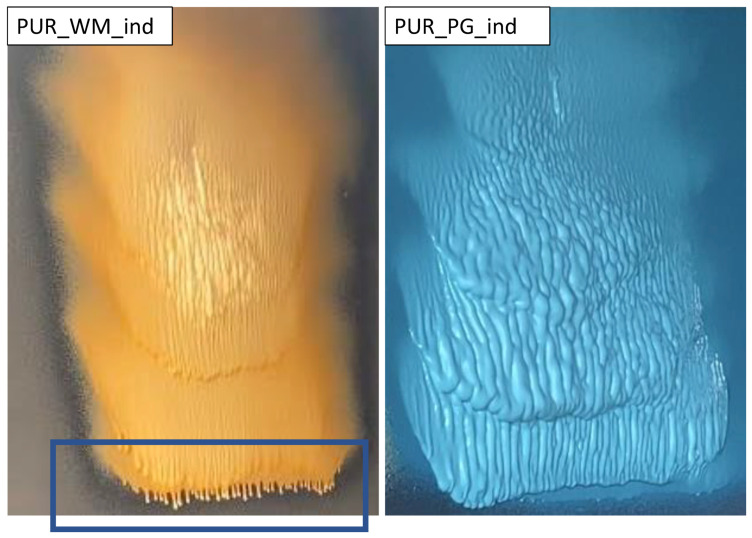
Open-cell PUR foam systems 5 s after spraying onto a vertical surface (the system flowing from the wall is marked with a blue frame).

**Table 1 polymers-16-01145-t001:** Formulations of open-cell spray PUR foam systems based on different biopolyols.

Symbol	PUR_BR_lab PUR_BR_ind, g	PUR_CH_lab PUR_CH_ind, g	PUR_GR_lab PUR_GR_ind, g	PUR_PG_lab PUR_PG_ind, g	PUR_WM_lab PUR_WM_ind, g
Biopolyol BR	100				
Biopolyol CH		100			
Biopolyol GR			100		
Biopolyol PG				100	
Biopolyol WM					100
Polycat 142	4	4	4	4	4
Tegostab 8870	2	2	2	2	2
Tegostab 8523	0.3	0.3	0.3	0.3	0.3
Ortegol 500	1	1	1	1	1
TCPP	20	20	20	20	20
Water	20	20	20	20	20
Isocyanate index	0.5	0.5	0.5	0.5	0.5

**Table 2 polymers-16-01145-t002:** Dimensional stability and oxygen index of foams measured at −25 and 70 °C.

Symbol	Dimensional Stability at −25 °C, %			Dimensional Stability at +70 °C, %			Oxygen Index, %
	DS_l_	DS_w_	DS_t_	DS_l_	DS_w_	DS_t_	
PUR_BC_lab	0.10	−0.02	−0.36	−0.09	−0.25	−0.17	21.0 ± 0.1
PUR_CH_lab	0.09	0.02	0.61	−0.50	−0.47	0.11	21.0 ± 0.1
PUR_GR_lab	0.24	0.24	0.53	−0.04	−0.08	0.09	20.6 ± 0.1
PUR_PG_lab	−0.76	−0.24	−0.06	−0.24	−0.63	0.74	20.9 ± 0.1
PUR_WM_lab	−0.10	−0.03	−0.72	−0.47	−0.35	−0.09	20.7 ± 0.1
PUR_BC_ind	−0.59	0.04	0.34	−0.19	−0.16	0.22	21.4 ± 0.1
PUR_CH_ind	0.64	0.06	−0.31	−0.20	−0.12	−0.21	21.3 ± 0.1
PUR_GR_ind	0.80	0.32	0.05	−0.19	0.06	0.28	21.4 ± 0.1
PUR_PG_ind	−0.17	−0.05	−0.45	−0.38	−0.41	−0.14	21.2 ± 0.1
PUR_WM_ind	−0.03	−0.09	−0.19	−0.07	−0.26	−0.03	21.3 ± 0.1

## Data Availability

Data are contained within the article.

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
