# Peer review of "Open-Cell Spray Polyurethane Foams Based on Biopolyols from Fruit Seed Oils"

_polymers, 2024, doi:10.3390/polym16081145_

Round 1

Reviewer 1 Report

Comments and Suggestions for Authors

The work is devoted to the elaboration of open-cell polyurethane foams (PUFs) based on the biopolyols. The employment of the biopolyols from renewable raw materials allows one to replace petrochemical polyols and, thus, decrease the emission of greenhouse gases. The authors present results on the replacement of synthetic polyols with blackcurrant, cherry, grape, pomegranate and watermelon seed oils. The structure, mechanical properties as well as fire resistance of the obtained foams were investigated.

The manuscript is well structured and written in logic and clear way. The choice of the analytical equipment is adequate. However, the work cannot be published in Polymers in its present state since it strongly lacks discussion part. Now, the authors just present the obtained results with no reference to the previous results. It would be nice to find a discussion part containing for example a comparison of the obtained mechanical characteristics or/and fire resistance with the same parameters of other PUFs previously synthesized with use of biopolyols (maybe some tables or diagrams). Moreover there are some issues listed below:

1) Lines 85-87: I did not catch the sense of the phrase.

2) Table 1: what are the units: %, grams or something else? Where is isocyanate in the table?

3) What was the sense of measuring of closed-cell content if this information is not presented in results (only in abstract)?

4) It would be nice to see more discussion on the reasons why PUR_WM_ind has the highest density?

5) There are two "Figure 3" in the manuscript, should be corrected.

6) Actually the apparent density and cell density could be presented in one figure.

7) Could you be so kind to present the values of LOI in Table, please? The Figure 7 is not too informative. Why LOIs of industrial foams are higher than the same values of laboratory foams? Density?

8) The conclusions should be extended.

Author Response

Dear Reviewer,

The authors would like to thank you for your time and the review of the article. Your comments helped us improve this manuscript. We corrected the manuscript according to your comments and/or include suitable explanations. All the changes in the text are marked with red font. We hope that the corrected paper will be suitable for publication in the Polymers. Below please find our answers to your comments.

Yours sincerely

Maria Kurańska et al.

The work is devoted to the elaboration of open-cell polyurethane foams (PUFs) based on the biopolyols. The employment of the biopolyols from renewable raw materials allows one to replace petrochemical polyols and, thus, decrease the emission of greenhouse gases. The authors present results on the replacement of synthetic polyols with blackcurrant, cherry, grape, pomegranate and watermelon seed oils. The structure, mechanical properties as well as fire resistance of the obtained foams were investigated.

The manuscript is well structured and written in logic and clear way. The choice of the analytical equipment is adequate. However, the work cannot be published in Polymers in its present state since it strongly lacks discussion part. Now, the authors just present the obtained results with no reference to the previous results. It would be nice to find a discussion part containing for example a comparison of the obtained mechanical characteristics or/and fire resistance with the same parameters of other PUFs previously synthesized with use of biopolyols (maybe some tables or diagrams). Moreover there are some issues listed below:

The research results obtained in this study were compared with previously published results regarding open-cell polyurethane foams modified with biopolyols.

  • Lines 85-87: I did not catch the sense of the phrase.

This part has been corrected.

  • Table 1: what are the units: %, grams or something else? Where is isocyanate in the table?

The unit has been added. The isocyanate index is given in the last row in table 1. It is 0.5.

  • What was the sense of measuring of closed-cell content if this information is not presented in results (only in abstract)?

The content of closed cells has been added to the cell density graph.

  • It would be nice to see more discussion on the reasons why PUR_WM_ind has the highest density?

The difference of approximately 2 kg/m3 is not very big. This may be caused by the method of spraying. It is not easy to prepare such small samples for testing using a high-pressure device. On a laboratory scale, where we are able to completely control the repeatability, these differences are much smaller.

5) There are two "Figure 3" in the manuscript, should be corrected.

It has been corrected

6) Actually the apparent density and cell density could be presented in one figure.

The cell density chart includes data on the content of closed cells. The authors decided not to show the apparent density in this figure because the graph might be unreadable.

7) Could you be so kind to present the values of LOI in Table, please? The Figure 7 is not too informative. Why LOIs of industrial foams are higher than the same values of laboratory foams? Density?

According to the Reviewer's comment, the oxygen index value is presented in the table 4.

8) The conclusions should be extended.

This part has been completed.

Reviewer 2 Report

Comments and Suggestions for Authors

Biopolyols were synthesis using the transesterification reaction by natural oils from watermelon, cherry, black currant, grape and pomegranate fruit seeds. the preparation possibility of open-cell foams from polyurethane system in which petrochemical polyol was fully replaced with biopolyols is analyzed. This research is a very interesting. Moreover, the author conducted a detailed analysis of the performance. But I think it still need minor revise before publication.

(1) The author's explanation of the mechanism of material performance analysis is insufficient. If necessary, some references can be cited to explain the mechanism.

(2) The model and testing method of optical microscope should be added.

(3) The instrument of limiting oxygen index (LOI) test should be added.

(4) The method for determining the hydroxyl numbers of the biopolyols require.

Comments on the Quality of English Language

Extensive editing of English language required

Author Response

Dear Reviewer,

The authors would like to thank you for your time and the review of the article. Your comments helped us improve this manuscript. We corrected the manuscript according to your comments and/or include suitable explanations. All the changes in the text are marked with red font. We hope that the corrected paper will be suitable for publication in the Polymers. Below please find our answers to your comments.

Yours sincerely

Maria Kurańska et al.

Biopolyols were synthesis using the transesterification reaction by natural oils from watermelon, cherry, black currant, grape and pomegranate fruit seeds. the preparation possibility of open-cell foams from polyurethane system in which petrochemical polyol was fully replaced with biopolyols is analyzed. This research is a very interesting. Moreover, the author conducted a detailed analysis of the performance. But I think it still need minor revise before publication.

  • The author's explanation of the mechanism of material performance analysis is insufficient. If necessary, some references can be cited to explain the mechanism.

The research results obtained in this study were compared with previously published results regarding open-cell polyurethane foams modified with biopolyols.

(2) The model and testing method of optical microscope should be added.

It has been added.

(3) The instrument of limiting oxygen index (LOI) test should be added.

It has been added.

(4) The method for determining the hydroxyl numbers of the biopolyols require.

It has been added.

Round 2

Reviewer 1 Report

Comments and Suggestions for Authors

The authors partially revised the manuscript and corrected some issues. For instance, the authors have added a comparison of the obtained experimental data with literature ones.

Nevertheless, the main drawback of the work (absence of any discussion on the obtained results) is still there. For example, if one really considers that one day the bio-polyols may replace the synthetic ones, which bio-polyol would the best option according to the authors findings? Why? It would nice to see an attempt to find any correlation between foam/cell densities and compressive strengths/LOI values. 

Unfortunately, without discussion I cannot recommend the article to publication.  

Author Response

Thank you for your comments. We have added explanations that we hope will be satisfactory. Best regards. Changes are marked in blue.

Best regards

Maria Kurańska

The authors partially revised the manuscript and corrected some issues. For instance, the authors have added a comparison of the obtained experimental data with literature ones.

Nevertheless, the main drawback of the work (absence of any discussion on the obtained results) is still there. For example, if one really considers that one day the bio-polyols may replace the synthetic ones, which bio-polyol would the best option according to the authors' findings? Why? It would be nice to see an attempt to find any correlation between foam/cell densities and compressive strengths/LOI values.

The additional explanations have been added.

Round 3

Reviewer 1 Report

Comments and Suggestions for Authors

The work can be published in its present form.